# T2A-Feedback: Improving Basic Capabilities of Text-to-Audio Generation via Fine-grained AI Feedback

Suddenly, a **man shouted, "Fire!"** The man and women joined in. **Two children cried together**. In no time, **thousands of people were shouting, thousands of children were crying, and countless dogs were barking**. Amid the chaos, there were sounds of **collapsing buildings, explosions, and strong winds**, all happening at once. There were also **cries for help,** the sounds of **buildings being dragged,** voices of **looting,** and **water splashing** everywhere.
（忽一人大呼："火起"，夫起大呼，妇亦起大呼。两儿齐哭。俄而百千人大呼，百千儿哭，百千犬吠。中间力拉崩倒之声，火爆声，呼呼风声，百千齐作；又夹百千求救声，曳屋许许声，抢夺声，泼水声。）

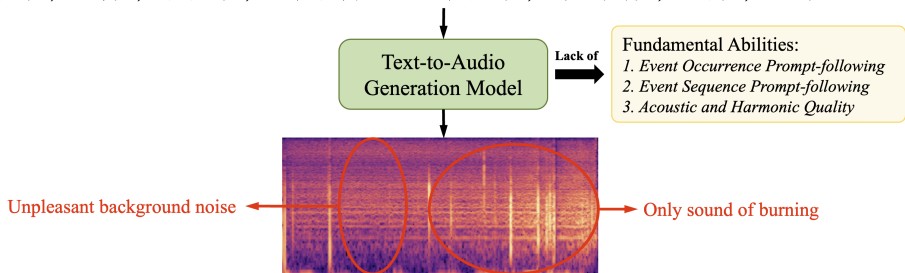

Figure 1: The audio description is from a classic Chinese essay "Kou Ji", which vividly depicts a performer using only vocal mimicry to recreate an entire dramatic scene. The existing Text-to-Audio generation model struggles to generate such narrative and multi-event audios. The generated audio often fails to contain all events in the described sequence while maintaining acoustic quality and harmony.

## ABSTRACT

Text-to-audio (T2A) generation has achieved remarkable progress in generating a variety of audio outputs from language prompts. However, current state-of-the-art T2A models still struggle to satisfy human preferences for prompt-following and acoustic quality when generating complex multi-event audio. To improve the performance of the model in these high-level applications, we propose to enhance the basic capabilities of the model with AI feedback learning. First, we introduce fine-grained AI audio scoring pipelines to: 1) verify whether each event in the text prompt is present in the audio (**Event Occurrence Score**), 2) detect deviations in event sequences from the language description (**Event Sequence Score**), and 3) assess the overall acoustic and harmonic quality of the generated audio (**Acoustic Harmonic Quality**). We evaluate these three automatic scoring pipelines and find that they correlate significantly better with human preferences than other evaluation metrics. This highlights their value as both feedback signals and evaluation metrics. Utilizing our robust scoring pipelines, we construct a large audio preference dataset, **T2A-FeedBack**, which contains 41k prompts and 249k audios, each accompanied by detailed scores. Moreover, we introduce **T2A-EpicBench**, a benchmark that focuses on long captions, multi-events, and story-telling scenarios, aiming to evaluate the advanced capabilities of T2A models. Finally, we demonstrate how T2A-FeedBack can enhance current state-of-the-art audio model. With simple preference tuning, the audio generation model exhibits significant improvements in both simple (AudioCaps test set) and complex (T2A-EpicBench) scenarios. The project page is available at https://T2Afeedback.github.io

## 1 INTRODUCTION

Recent Text-to-Audio (T2A) generation models (Huang et al., 2023b;a; Liu et al., 2023a; 2024; Ghosal et al., 2023; Majumder et al., 2024; Vyas et al., 2023) have made drastic performance improvements. By trained on massive audio-text data (Gemmeke et al., 2017; Fonseca et al., 2021; Chen et al., 2020; Kim et al., 2019), these generative models learn to generate diverse sounds with a given language prompt. For audio generation, generating harmonious multi-event audio or describing a story with audio has important applications in music (Agostinelli et al., 2023), advertising, video-audio generation (Luo et al., 2024; Wang et al., 2024), etc. However, as shown in Figure. 1, existing audio generation models are struggling to generate harmonious and high-quality audio from narrative and complex descriptions, which limits the potential for high-level applications.

The failure of the generated results is often demonstrated in three aspects: 1) cannot fully include all the events described, 2) cannot accurately follow the order of all the events described, and 3) cannot organize all the events harmoniously. Therefore, the model performance in multi-event scenarios is determined by its capabilities in these three fundamental aspects.

To improve the model's performance across more advanced applications, we focus on strengthening the audio generation model's fundamental abilities. Inspired by feedback learning in large language models (Ouyang et al., 2022; Bai et al., 2022; Touvron et al., 2023), we propose creating an audio preference dataset centered on three abilities necessary for generating harmonic and complex audio: 1) **Event Occurrence Prompt-Following**, 2) **Event Sequence Prompt-Following**, and 3) **Acoustic&Harmonic Quality**. Based on the preference information, we can refine the model's core abilities, resulting in better results in both simple and challenging scenarios.

However, due to the scarcity of audio data and the challenges of annotating the scale of user preferences, it is difficult to collect massive audio preference datasets that only rely on human annotators. To fill this void, we explore using AI feedback (Cui et al., 2023; Lee et al., 2023; Yuan et al., 2024; Burns et al., 2023) in text-to-audio generation, utilizing AI models to rank audios instead of relying on human annotators. Compared to manual annotation, automating the data collection and annotation process reduces the cost of obtaining audio preference data and enhances scalability.

Specifically, we develop three AI scoring pipelines to evaluate the generated audio in a fine-grained and holistic manner, corresponding to three core capabilities:

- *Event Occurrence Score*: To specifically check whether each event occurs in, we calculate the audio-text semantic matching score for each described event separately. A lower score suggests that the corresponding event might be absent from the audio.

- *Event Sequence Score*: To verify the correctness of event order, we analyze the start and end times of each event and compare them with the event order outlined in the text prompt. A higher score implies a greater similarity between the event sequences in caption and audio.

- *Acoustics&Harmonic Quality*: Drawing inspiration from aesthetic scoring methods used in image quality scoring, we manually annotate acoustic and harmonic quality for audio samples. These data are then used to train an automatic acoustic&harmonic predictor.

We confirm that our three scoring functions show a stronger correlation with human evaluations compared to existing automatic audio evaluation methods (Wu et al., 2023b; Xie et al., 2024). Consequently, in addition to their application in ranking preference data, these scoring functions can be used as evaluation metrics that more effectively capture human preferences across different aspects.

Leveraging these advanced AI scoring pipelines, we establish a comprehensive data collection and annotation framework. As a result, we construct **T2A-Feedback**, a large audio preference dataset comprising 41,627 captions and 249,762 generated audios, each annotated with detailed scores.

Furthermore, to evaluate the higher-level capabilities of text-to-audio models in multi-event scenarios, we introduce a more challenging benchmark, **T2A-EpicBench**, which features longer, more imaginative, and story-telling captions for audio generation. We enhance the advanced text-to-audio diffusion model, Make-an-Audio 2 (Huang et al., 2023a), with T2A-Feedback. Our results show that using T2A-Feedback not only effectively improves the basic capabilities of the model in simple AudioCaps benchmark, but also emergently improves the performance in complex T2A-EpicBench.

## 2 RELATED WORK

### 2.1 TEXT-TO-AUDIO GENERATION

Text-to-audio generation is an emerging field that aims to convert textual descriptions into corresponding audio outputs. Existing text-to-audio generation methods can be divided into two categories: Diffusion-based and Language model-based. Diffusion-based techniques have gained prominence for generating high-quality, realistic audio by modeling the process of denoising. These methods, like Make-an-Audio (Huang et al., 2023b;a), AudioLDM (Liu et al., 2023a; 2024), Tango (Ghosal et al., 2023; Majumder et al., 2024), start with random noise and iteratively refine it to produce coherent audio over a series of denoising steps. On the other hand, Language model-based methods (Borsos et al., 2023; Agostinelli et al., 2023; Cideron et al., 2024) tokenize audios as acoustic discrete tokens, and predict the tokens within an auto-regressive model conditioned on text inputs.

The above models acquire the ability to generate diverse audio by training on large-scale audio-text datasets. However, current datasets like AudioSet (Gemmeke et al., 2017), AudioCaps (Kim et al., 2019), and FSD50k (Fonseca et al., 2021) only provide tag-level annotations or short captions. As a result, when processing long, detailed language prompts, existing models often produce low-quality, noisy outputs and struggle to accurately follow the text. Due to the difficulty of annotating detailed audio captions, scaling rich and accurate audio descriptions remains a challenge. In this work, we focus on enhancing the model's basic abilities in event occurrence, sequence, and harmony, thereby improving its performance in both simple scenarios and advanced applications.

### 2.2 PERFERENCE TUNING WITH HUMAN&AI FEEDBACK

Tuning generative models according to human preferences has emerged as a standard practice for improving the quality of outputs. By tuning with feedback information on different aspects, the model can be improved and aligned with human preferences in corresponding aspects. Traditionally, this preference data used for tuning relied heavily on human evaluators who rank multiple generated results, assessing their quality based on various criteria such as relevance, coherence, and aesthetic value (Bai et al., 2022; Touvron et al., 2023; Ouyang et al., 2022; Kirstain et al., 2023; Liang et al., 2024; Wu et al., 2023a; Cideron et al., 2024).

While effective, manual human annotation is costly and time-consuming, which greatly hampers the scalability of preference tuning across more diverse generative tasks. To address the difficulty, more recent developments have focused on leveraging pre-trained AI models to automate the process of scoring generated content (Cui et al., 2023; Lee et al., 2023; Yuan et al., 2024; Burns et al., 2023). Such an AI feedback approach has achieved impressive improvements in large language models.

Recently, some studies have attempted preference fine-tuning in text-to-audio generation models. One recent paper related to our work, Tango2 (Majumder et al., 2024), utilizes contrastive language-audio pre-training (CLAP) (Wu et al., 2023b) to rank audio generated by the Tango model. However, CLAP can only evaluate the global alignment between audio and text but falls short in assessing the fine-grained details, like detailed event occurrence, sequence, and harmony. In this paper, we construct more robust AI audio scoring pipelines with fine-grained recognition ability. Our method shows a much stronger correlation with human preference and the constructed dataset brings significant improvement to the current text-to-audio generation model.

### 2.3 TEXT-TO-AUDIO EVALUATION METRIC

Existing evaluation metrics for audio generation, such as FAD and IS, assess audio distributions but cannot evaluate the quality of individual samples. Additionally, many studies rely on similarity scores from the CLAP model to assess global audio-text semantic alignment. PicoAudio (Xie et al., 2024) uses a text-to-audio grounding model (Xu et al., 2024) to detect audio segments based on language prompts. However, there remains a lack of fine-grained evaluation methods for assessing detailed event occurrence, sequencing, and acoustic quality. Our research fills this gap by creating robust audio AI scoring pipelines, that show a strong correlation with humans, and significantly surpass alternative methods.

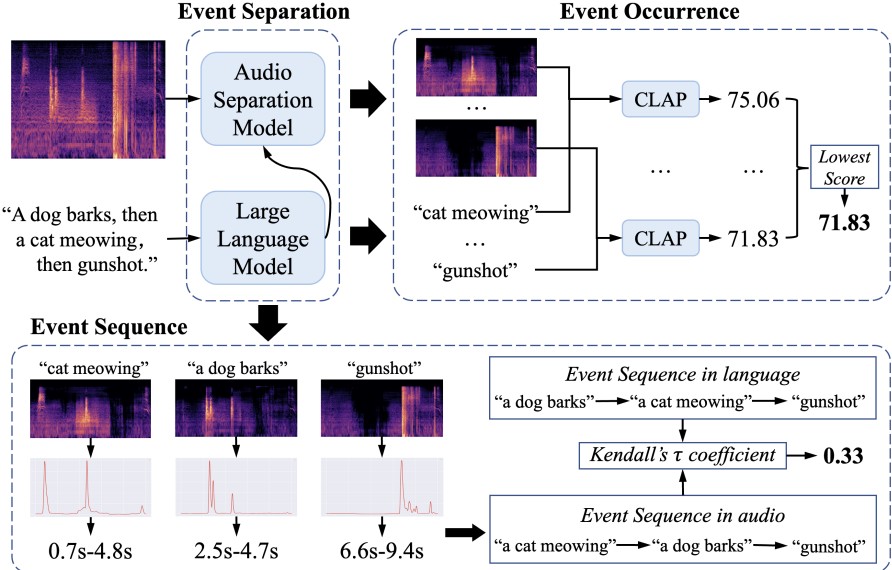

Figure 2: The overview of event occurrence and sequence scoring pipelines.

## 3    T2A-FEEDBACK

In this section, we first dive into the three AI audio scoring pipelines: (i) Event Occurrence Prompt-following, in Section. 3.1; (ii) Event Sequence Prompt-following, in Section. 3.2; (iii) Acoustic Quality, in Section. 3.3. We then describe the specific data generation and sorting method for the T2A-Feedback dataset in Section. 3.4.

### 3.1    EVENTS OCCURRENCE PROMPT-FOLLOWING

Generating audio that accurately reflects the events described in a given prompt is the fundamental requirement of prompt-following. However, when multiple events are included in the text description, current text-to-audio generation models often struggle to generate each event precisely. To improve the generation model's event occurrence prompt-following ability, we first build an AI pipeline to determine the occurrence of events in audio.

Previous methods primarily utilize contrastive language-audio pre-training (CLAP) (Wu et al., 2023b) over the audios and language descriptions to assess their semantic relevance. However, in multi-event scenarios, the sentence-level matching score struggles to identify event-level misalignment, and can not pinpoint which specific events are present and which are not, as shown in Figure. 5. To accurately identify misaligned events, we propose to measure the audio-text semantic alignment at the event-level. To this end, we first separate the language description and audio into basic events, as shown in the "Event Separation" part of Figure. 2. Specifically, we utilize a large language model (LLM) (Jiang et al., 2023) to decompose descriptions into event captions according to the described order. Meanwhile, we employ an advanced audio separation model (Liu et al., 2023b) to segment the audio into event-level sub-audios based on these event captions. By calculating the similarity between each event-level description and its corresponding sub-audio in CLAP space, we can gain clearer insights into the specific aligned and misaligned events.

To encourage the models to comprehensively generate all described events, for each audio-text pair, we select the lowest value among all event-level audio-text matching scores as the **Event Occurrence Score**. For audios generated from the same caption, a higher score indicates that the audio is more likely to contain all the described events.

### 3.2    EVENTS SEQUENCE PROMPT-FOLLOWING

In addition to generating all events, whether these events occur in the temporal order described in the caption is also a crucial aspect of prompt-following. Some recent work attempts to detect the sequence of events in audio. Tango2 (Majumder et al., 2024) computes the CLAP matching score between the temporal description and corresponding audios, but we find the sentence-level

CLAP score is not sensitive to the temporal description in captions, as demonstrated in Figure. 5 and Table. 2. On the other hand, PicoAudio (Xie et al., 2024) employs audio grounding model (Xu et al., 2024) to detect audio segments. However, due to the limitation of the training scale, the generalization performance of the audio grounding model is limited.

To robustly analyze audio event sequences, we propose a new pipeline for event sequence analysis. Similar to event occurrence, we first use the LLM and audio separation model to extract event-level descriptions and their corresponding sub-audios. For each separated audio track, we determine the event's start and end times based on volume levels. Specifically, we normalize the volume to a range of [0,1], and the period where the normalized volume exceeds a certain threshold is identified as the event's duration.

In multi-event scenarios, there are multiple complex temporal relationships. To comprehensively assess the temporal alignment between the language prompt and the generated audio, and to specifically identify which temporal relationships are accurate and which are misaligned, we employ Kendall's $\tau$ coefficient. This widely used non-parametric statistic measures rank correlation between two variables. Considering $n$ events and their $n(n-1)$ event pairs, we use LLM to analyze the relationships between each event pair in the language description and extract the event sequence in the audio based on the starting time of each event. The **Events Sequence Score** (e.g., Kendall's $\tau$ coefficient between event sequences in language and audio) is calculated as:

$$\tau = \frac{C - D}{n(n-1)} \tag{1}$$

where $C$ represents the number of concordant event pairs between the description and the audio, $D$ denotes the number of discordant ones. Higher $\tau$ indicates a greater alignment of the event sequence in the generated audio with the text description. Specifically, $\tau = 1$ signifies that the event sequence in the generated audio is identical to the language description, while $\tau = -1$ indicates that the sequences are completely reversed.

## 3.3 ACOUSTIC&HARMONIC QUALITY

In addition to generating all events accurately following the language prompt, organically integrating different events to create a pleasant-sounding effect is also one of the basic capabilities. However, current audio generation models often produce low-quality and noisy results.

To alleviate this challenge, we first develop an audio acoustic&harmonic quality predictor. Inspired by the image aesthetic predictor in Schuhmann et al. (2022), we first manually score audio samples on a scale from 1 to 4 according to their quality. Two annotators independently score the audio samples according to the same criteria, and samples with consistent scores are accepted as training data. The detailed scoring criteria are as follows:

Annotators need to score the auditory quality of audio from the following four perspectives:
**Acoustic Quality**: Does the generated audio sound realistic and pleasant?
**Harmony**: Do different sound elements integrate well, forming a cohesive auditory scene?
**Background Noise**: Is there noise that disrupts the clarity and naturalness of the audio?
**Dynamic Range**: Are the different audio elements within their reasonable volume range?
The specific standards for each score are as follows:

| Score | Standard |
| --- | --- |
| 1 | Poor audio quality; sounds unrealistic with disjointed elements, severe background noise interference, and extremely limited dynamic range. |
| 2 | Normal audio quality; some events are natural, but overall harmony is lacking. Background noise affects clarity, and dynamic range is limited. |
| 3 | Good audio quality; most events are realistic with harmonious integration. Background noise is minimally disruptive, and dynamic range is reasonable. |
| 4 | Excellent audio quality; all events are very realistic with perfect integration, well-managed background noise, and wide dynamic range. |

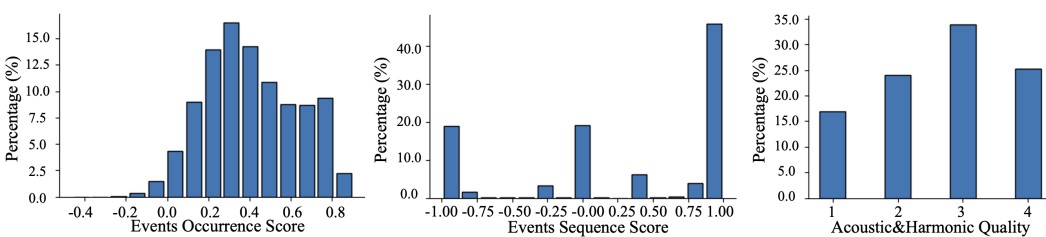

Figure 3: Histograms of three different scores in T2A-Feedback.

Using the human-annotated data, we train a linear predictor on the top of CLAP audio embeddings. With the high-quality pre-trained representation, we find that, akin to aesthetic score predictors for images, a small amount of annotated data can yield a generalized subjective quality predictor. Specifically, we train the acoustic predictor with 1,000 meticulously annotated audio samples using cross-entropy loss. The output of the predictor is termed the **Acoustic&Harmonic Quality**.

### 3.4 PREFERENCE DATA GENERATION

To generate diverse and comprehensive audio, we first augment the text prompts used for audio generation. We begin with the captions from the training set of the large-scale audio-text dataset, AudioCaps. By employing an LLM, we decompose these captions into fundamental event descriptions and calculate their semantic similarity within the CLAP space to filter out non-overlapping, basic event descriptions. Then, we prompt the LLM with randomly selected events to create varied and natural multi-event audio descriptions, with explicit temporal ordering. Finally, we combine the enhanced 3,769 captions with the 37,858 captions from the training set of AudioCaps, serving as the prompt source for audio generation.

As highlighted in Cui et al. (2023), diversity is crucial for preference datasets. To mitigate the potential bias of using a single audio generation model and to enhance the generalization of the generated data, we employ three advanced audio generation models: Make-an-Audio2, AudioLDM2, and Tango2. Each model generates 2 audio per caption, resulting in a total of 6 audio files for each caption. In summary, we produce 249,762 audios from 41,627 descriptions. For audios generated from the same captions, we combine three rankings of each score to derive the overall ranking.

The histogram plots of the scores on all the generated audios are shown in Figure. 3. The distribution of Event Occurrence Scores and Acoustic&Harmonic Quality is similar to a Gaussian distribution. Since most descriptions contain one or two sequential events, Event Sequence Scores are concentrated between -1 and 1. As noted in Liang et al. (2024), this discriminative score distribution ensures a balanced ratio of negative to positive samples, enabling effective preference tuning.

## 4  T2A-EPICBENCH

Current text-to-audio generation models are mainly evaluated and compared on the AudioCaps test set. However, the captions in AudioCaps are generally short and simple, averaging 10.3 words per sentence. Specifically, 17% of the captions feature only a single event, and 44% contains two events. This is not enough to assess the model's capabilities in more advanced applications involving detailed, multi-event, and narrative-style audio generation.

To fill this gap, we propose **T2A-EpicBench**, consisting of 100 detailed, multi-event, and story-telling captions. Each caption averages 54.8 words and 4.2 events, with 86% containing four events and the remainder featuring five or more. Initially, we manually write 10 detailed captions, then used them as in-context examples to prompt LLM for generating the remaining captions. All 100 captions are manually reviewed for accuracy. Several examples from T2A-EpicBench are included in the Appendix.

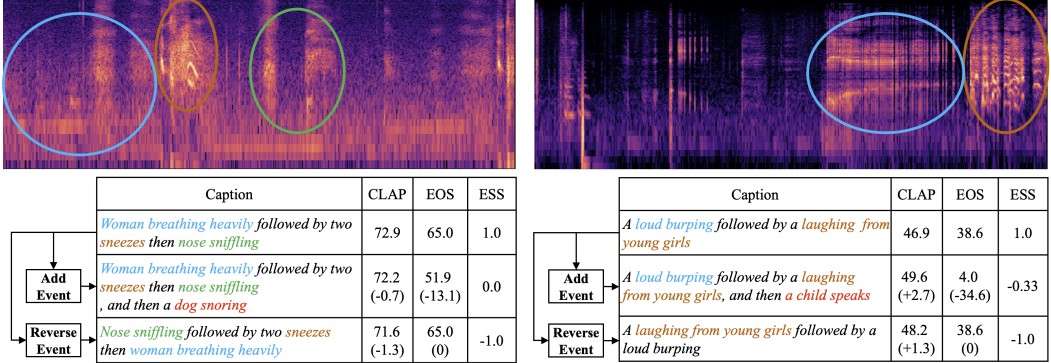

(a) *Pigeons cooing as air lightly hisses in the background followed by a camera muffling*

EOS: 35.30   ESS: 1.0   AHQ: 4

(b) *Loud wind noise followed by a car accelerating fast, and then water softly trickling*

EOS: -10.17   ESS: 0.0   AHQ: 4

(c) *Toilet flushing followed by door closes*

EOS: 23.39   ESS: -1.0   AHQ: 4

(d) *Digital beeps followed by static electric hissing*

EOS: 23.40   ESS: 1.0   AHQ : 1

Figure 4: Visualization of the predicted scores from our AI scoring pipeline. We highlight the first, second, and third events described in the captions using blue, brown, and green, respectively.

| | Caption | CLAP | EOS | ESS |
|---|---|---|---|---|
| | *Woman breathing heavily followed by two sneezes then nose sniffling* | 72.9 | 65.0 | 1.0 |
| Add Event | *Woman breathing heavily followed by two sneezes then nose sniffling, and then a dog snoring* | 72.2 (-0.7) | 51.9 (-13.1) | 0.0 |
| Reverse Event | *Nose sniffling followed by two sneezes then woman breathing heavily* | 71.6 (-1.3) | 65.0 (0) | -1.0 |

| | Caption | CLAP | EOS | ESS |
|---|---|---|---|---|
| | *A loud burping followed by a laughing from young girls* | 46.9 | 38.6 | 1.0 |
| Add Event | *A loud burping followed by a laughing from young girls, and then a child speaks* | 49.6 (+2.7) | 4.0 (-34.6) | -0.33 |
| Reverse Event | *A laughing from young girls followed by a loud burping* | 48.2 (+1.3) | 38.6 (0) | -1.0 |

Figure 5: Qualitative comparison between CLAP scores and EOS/ESS scores reveals distinct sensitivities to misalignment. By adding or reversing events in the ground-truth caption, we create captions that are misaligned with the audio in terms of event occurrence and sequence.

# 5 EXPERIMENT

## 5.1 ANALYSIS OF AI SCORING PIPELINES

### 5.1.1 QUANTITATIVE ANALYSIS

**Evaluation of Event Occurrence Score (EOS)** To evaluate the scoring model's capability in recognizing whether audios contain all the events described in the text, we propose a missing event recognition task. We construct distracting captions for the AudioCaps test set, by adding random event descriptions to the ground-truth captions. This task challenges models to distinguish the ground-truth caption from the constructed interference captions. There are 3,701 samples in total for this task.

Table 1: Comparison about event occurrence

| | Accuracy |
|---|---|
| Random Guess | 50.0% |
| CLAP | 77.5% |
| EOS(ours) | **90.9%** |

We mainly compare our EOS with sentence-level CLAP score. The caption with the higher matching score to the audio is considered as the prediction. As shown in Table 1, our EOS score showcases a notable advantage over CLAP, demonstrating the superiority of event-level audio-text matching in identifying whether all events are correctly contained in audios.

**Evaluation of Event Sequence Score (ESS)** To verify the ability to distinguish the alignment of event sequences in text and audio, we collect 450 two-event samples from PicoAudio's training set, and reverse the events orders in the description as interference caption. Using this dataset, we compare different methods by calculating the accuracy of recognizing the ground-truth description versus the interference description, and by evaluat-

Table 2: Comparison about event sequence. $ESS_{0.x}$ stands for using $0.x$ as volume thresholds.

| Method | Accuracy | F1 Score | Correlation |
|---|---|---|---|
| CLAP | 49.6 | - | - |
| PicoAudio | 71.6 | 0.787 | 0.30 |
| $ESS_{0.1}$ | **79.6** | 0.814 | 0.43 |
| $ESS_{0.3}$ | 79.1 | **0.851** | **0.52** |
| $ESS_{0.5}$ | 78.0 | 0.769 | 0.52 |

ing the Segment F1 Score (Mesaros et al., 2016) for detecting the start and end times of each audio event. Moreover, we manually annotate temporal order alignment for 100 audios generated from our temporal-enhanced captions and compute the correlation between different methods and humans.

The results of event sequences are provided in Table. 2. We compare ESS with CLAP score and the audio grounding model (Xu et al., 2024) used by PicoAudio (Xie et al., 2024). Compared to baselines, our method distinguishes the ground-truth caption from the distracting one more accurately and achieves higher F1 scores in detecting the start and end times of events in audio. More importantly, our method shows a much stronger correlation to human annotations.

Additionally, we investigate various volume thresholds used to determine the duration of each event. In Table 2, we test thresholds of 0.1, 0.3, and 0.5. ESS consistently performs better than other methods across most settings, with 0.3 providing the optimal results and thus chosen as the default setting.

**Evaluation of Acoustic&Harmonic Quality (AHQ)**  To validate our acoustic&harmonic predictor, we independently annotate 100 additional audios as a test set. The correlation between the model predictions and human labels on the test set is 0.786, showing strong generalization ability and high consistency with human preferences.

Moreover, we explore building the Acoustic&Harmonic Predictor on top of various pre-trained audio models and evaluate how well each variant correlates with human preferences. The results in Table 3 show that the predictor built on CLAP (Wu et al., 2023b) outperforms those based on self-supervised models like AudioMAE (Huang et al., 2022) and BEAT (Chen et al., 2022). Similarly, the image aesthetics predictor (Schuhmann et al., 2022) is built on the CLIP model (Ilharco et al., 2021). This advantage may stem from the fact that self-supervised models are task-agnostic, whereas CLIP and CLAP align with language, resulting in better semantic discrimination.

Table 3: AHQ Predictor on different base models.

|  | Correlation |
| --- | --- |
| AudioMAE | 0.613 |
| BEAT | 0.519 |
| CLAP | **0.786** |

### 5.1.2 QUALITATIVE ANALYSIS

We show some example predictions from our scoring pipelines in Figure. 4, where our methods can specifically identify the misaligned event, the out-of-order event order, and the disharmony between events in the audio. Moreover, we provide the confusion matrix of acoustic&harmonic predictor on these 100 test samples in Figure. 6, which further demonstrates the statistical robustness of our predictor.

Moreover, we provide the qualitative comparison between our EOS and ESS with the single CLAP score, in Figure. 5. For the ground-truth audio-caption pairs from AudioCaps, we perturb the captions by adding an event or shuffling the order of events. We find that the CLAP score is not sensitive to these perturbations and even yields a

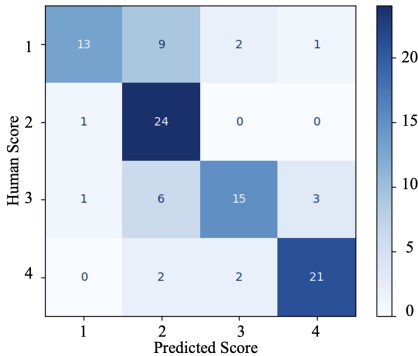

Figure 6: Confusion matrix.

higher score with the incorrect, perturbed caption. In contrast, our EOS and ESS scores more accurately reflect the alignment between audio and text regarding event occurrence and event order.

### 5.2 ANALYSIS OF PREFERENCE TUNING

To demonstrate the effect of T2A-Feedback dataset in improving audio generation model, we fine-tuning the advanced text-to-audio model, Make-an-Audio 2 (Huang et al., 2023a), with two preference training methods: Direct Preference Optimization (DPO) (Wallace et al., 2024) and Reward rAnked FineTuning (RAFT) (Dong et al., 2023). Another audio preference dataset, Audio-Alpace, proposed by Majumder et al. (2024) is the main baseline for comparison. Both the widely-used AudioCaps and the new T2A-EpicBench are used as benchmarks, corresponding to applications in simple and complex scenarios respectively.

Table 4: Evaluation results on AudioCaps. The EOS, ESS and AHQ represent the Event Occurrence Score, Event Sequence Score and Acoustic&Harmonic Quality, respectively.

| | | FAD↓ | KL↓ | IS↑ | CLAP↑ | EOS.↑ | ESS.↑ | AQ.↑ |
|---|---|---|---|---|---|---|---|---|
| Make an Audio 2 | | **1.82** | 1.44 | 10.03 | 69.97 | 42.05 | 0.53 | 2.33 |
| Preference Tuning | | | | | | | | |
| Audio-Alpaca | RAFT | 1.93 | **1.29** | 10.37 | 72.23 | 44.85 | 0.53 | 2.45 |
| | DPO | 3.20 | 1.24 | **12.27** | 72.36 | 44.42 | 0.55 | 2.14 |
| T2A-Feedback | RAFT | 2.29 | 1.33 | 11.66 | 73.10 | 45.53 | 0.51 | 2.50 |
| (ours) | DPO | 2.64 | 1.31 | 11.35 | **74.00** | **49.58** | **0.57** | **2.57** |

Table 5: Evaluation results on T2A-EpicBench. The $win_{EOS}$, $win_{ESS}$ and $win_{AHQ}$ represent the win rates of tuned models over the original model in terms of Event Occurrence, Event Sequence and Acoustic&Harmonic Quality, respectively.

| | | AI Scoring | | | Human Scoring | | |
|---|---|---|---|---|---|---|---|
| | | $win_{EOS}$ | $win_{ESS}$ | $win_{AHQ}$ | $win_{EOS}$ | $win_{ESS}$ | $win_{AHQ}$ |
| Make an Audio 2 | | - (14.21) | - (0.03) | - (1.96) | - | - | - |
| Preference Tuning | | | | | | | |
| Audio-Alpaca | RAFT | 53%(15.73) | 51%(0.04) | 42%(1.69) | 57% | 54% | 53% |
| | DPO | 55%(16.87) | 52%(0.03) | 49%(1.96) | 65% | **64%** | 59% |
| T2A-Feedback | RAFT | 52%(15.85) | 52%(0.05) | **54%(2.14)** | 61% | 57% | 61% |
| (ours) | DPO | **58%(19.96)** | **64%(0.13)** | 52%(2.10) | **68%** | 62% | **68%** |

### 5.2.1 QUANTITATIVE RESULTS ON AUDIOCAPS

The classical automated metrics (FAD, KL, IS, and CLAP), as well as our three new scores (EOS, ESS, and AHQ) are employed to quantitatively evaluate and compare different model variants.

The quantitative results are provided in Table. 4. FAD, KL, and IS assess audio fidelity by evaluating the distribution of the generated audio. For these metrics, both the preference dataset and training methods result in similar overall improvements. CLAP is commonly used to measure the semantic alignment between the input prompt and the generated audio. While both Audio-Alpaca and T2A-Feedback improve the CLAP score, T2A-Feedback yields greater gains.

Moreover, as analyzed in Section. 5.1.1, the proposed EOS and ESS are more accurate than CLAP in judging event occurrence and event sequence, and AHQ shows a strong correlation to human preference in acoustic and harmony. We calculate the three scores for different model variants to evaluate audio generation results more accurately and comprehensively. The significantly better results across these three metrics demonstrate that T2A-Feedback yields far greater improvements compared to Audio-Alpaca, and the DPO method outperforms RAFT in our setting.

### 5.2.2 QUANTITATIVE RESULTS ON T2A-EPICBENCH

Since there are no ground-truth audios for the long and story-telling text prompts in T2A-EpicBench, we primarily measure the win rate of preference-tuned models against the original model outputs across three key areas: event occurrence, event sequence, and acoustic & harmonic quality. In addition to scoring the generated audio with our AI pipeline, we conduct a user study where two human annotators evaluate and select the better output based on each criterion.

The results on T2A-EpicBench, are illustrated in Table. 5, indicate that Audio-Alpaca provides only marginal improvements in handling detailed captions and multi-event scenarios, whereas T2A-Feedback significantly and comprehensively enhances the model's performance.

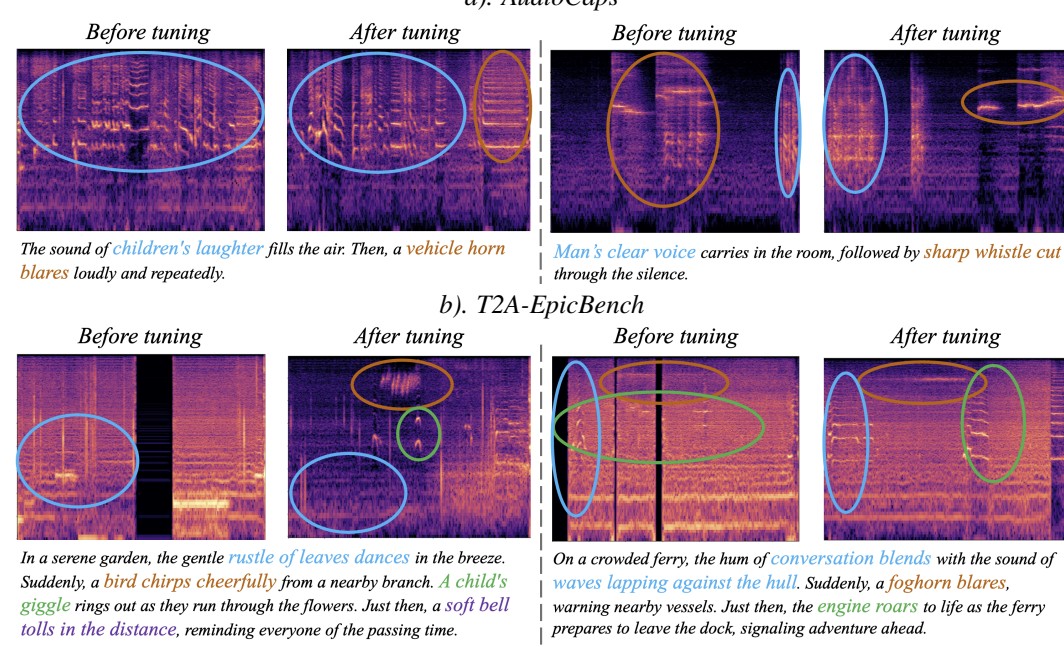

Figure 7: Visualization of the impact of preference tuning with T2A-Feedback.

It is worth noting that T2A-Feedback does not include long audio descriptions. The average word count per caption in T2A-Feedback is 9.6, which is considerably shorter than the 54.8 average word number of T2A-EpicBench prompts, and even shorter than Audio-Alpaca's 10.2 words per caption. T2A-Feedback does not directly provide additional long caption data, and the 65% average win rate in the user study reinforces that by focusing on improving the basic capabilities of short captions, the audio generation model can emergently learn to handle more complex long-text and multi-event scenarios.

### 5.2.3 QUALITATIVE FINDINGS

To better demonstrate the effectiveness of preference tuning on T2A-Feedback, we visualize some examples of tuning the original model on our T2A-Feedback with the DPO method in Figure. 7. For the examples of short captions in Figure. 7a, while both models before and after fine-tuning can produce clean audio, the fine-tuned model successfully generates all events in the described order. In the more challenging case from T2A-EpicBench, the original model often generates noisy, low-quality audio, making it difficult to distinguish the events. Preference tuning on T2A-Feedback, as shown in Figure. 7b, significantly reduces background noise and generates audio that more faithfully captures both events and their orders.

## 6 CONCLUSION

In this paper, we build AI scoring pipelines to evaluate three fundamental capabilities of audio generation: Event Occurrence Prompt-following, Event Sequence Prompt-following, and Acoustics&Harmonic Quality. Using these automatic evaluation metrics, which are highly correlated with human preferences, we build a large-scale audio preference dataset, **T2A-Feedback**. Experimentally, we extensively demonstrate the accuracy and robustness of our AI scoring pipelines. The three scores demonstrate a strong correlation to human preferences, which highlights its potential to better evaluate text-to-audio generation models. To assess the model's ability in complex multi-event scenarios, we propose a new challenging benchmark, **T2A-EpicBench**, which requires models to generate detailed and narrative audios. Using our T2A-Feedback to tune the audio generation model effectively improves its capabilities in the three core aspects and achieves better performance in both simple (AudioCaps) and complex (T2A-EpicBench) scenarios.

## REPRODUCIBILITY STATEMENT

The newly proposed audio AI scoring pipeline, preference dataset (T2A-Feedback) and benchmark (T2A-EpicBench) will be open-sourced. In addition, in Section 3, 5 and A, we describe our pipelines, evaluation tasks and data, and other implementation details in detail to ensure the reproducibility of our method.

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

## A    IMPLEMENTATION DETAILS

**Audio Generation**    During the audio generation process in T2A-Feedback, all models are set to 100 denoising steps with the DDIM scheduler, and classifier-free guidance is configured at 4.0.

**Training Details**    For Acoustic&Harmonic Predictor, we train an extra two-layer MLP projector on the top of CLAP audio representations using Cross Entropy(CE) loss. The predictor is trained using the Adam optimizer with a learning rate of 1e-2.5 for 6 epochs on 1,000 manually annotated data. For preference tuning, we employ the AdamW optimizer with a learning rate of 1e-5 for both DPO and RAFT strategy, and train one epoch for both Audio-Alpaca and T2A-Feedback.

## B    EXAMPLES FROM T2A-EPICBENCH

1.    At a lively beach, the waves crash rhythmically against the shore, providing a soothing melody. Suddenly, a seagull caws overhead, drawing attention from sunbathers. Children's giggles fill the air as they splash in the water. Just then, a distant drumbeat starts, adding a festive atmosphere to the scene.

2.    In a vibrant classroom, the teacher's voice resonates as she explains a lesson. Suddenly, a pencil rolls off a desk and clatters to the floor, causing a brief distraction. A student whispers a joke, provoking a wave of giggles. Just then, the school bell rings, signaling the end of the period and the excitement of freedom.

3.    In a busy city street, the honking of cars creates a chaotic symphony. Suddenly, a bicycle bell rings sharply as a cyclist weaves through traffic. The murmur of pedestrians chatting fills the air, blending with the distant sound of street performers playing music. Just then, the sound of footsteps approaches, adding to the urban rhythm.

4.    At a busy construction site, the sound of drills and saws fills the air, creating a symphony of labor. Suddenly, a heavy beam falls with a loud thud, causing workers to pause. A whistle blows, signaling a break, and conversations buzz among the crew. Just then, a truck backs up, beeping insistently as it arrives.

5.    In a vibrant downtown area, the honking of cars creates a chaotic symphony. Suddenly, a street vendor shouts out their specials, trying to attract customers. The laughter of people enjoying a nearby café adds warmth to the urban sounds. Just then, a bus rumbles past, its engine growling as it continues on its route.

6.    In a vibrant market, the chatter of vendors fills the air as they hawk their goods. Suddenly, a loud crash echoes as a stack of crates falls over, causing startled gasps. A nearby musician strums a guitar, trying to restore the upbeat mood. Just then, a child squeals with delight, tugging at their parent's hand to explore further.

7.    In a sunlit meadow, the buzzing of bees fills the air as they flit from flower to flower. Suddenly, a cow moos softly from a nearby barn, adding a pastoral charm. A couple of children giggle as they run through the grass, their joyful sounds mingling with nature. Just then, a breeze stirs, causing the wind chimes to tinkle gently.

8.    In a tranquil village square, the chirping of crickets fills the evening air. Suddenly, a family gathers around a fire pit, laughter and chatter rising in the dusk. The crackle of flames adds warmth to the scene. Just then, the distant call of an owl echoes, signaling the approach of night.

9.    In a dense forest, the soft rustle of leaves whispers through the trees as a gentle breeze blows. Suddenly, a twig snaps underfoot, startling a nearby deer, which bounds away with a soft thud. A bird sings a cheerful melody, filling the air with life. Just then, the distant sound of a waterfall cascades, creating a peaceful backdrop to the vibrant sounds of nature.

## C  LLM PROMPTS

The LLM used in Section 1 to separate basic events in the audio description, and in Section 2 for caption augmentation, is Mistral-7B-instruct-v0.2. The respective prompts are provided below:

messages = [ "role": "user", "content": "'I will provide a description of an audio, you need to break down the sentence and figure out the single sound elements. Use the words that appears in the sentence and appropriately replace the demonstrative pronouns if possible, such as it, she, him. The output should only include the decomposed sub-events. Here are some examples:
Description: A man speaks while ducks honk then birds vocalize. '",
"role": "assistant", "content": "Answers: (a man speaks while duck honk)@(birds vocalize)",
"role": "user", "content": "Description: Rain falls on a hard surface." ,
"role": "assistant", "content": "Answers: (rain falls on a hard surface)",
"role": "user", "content": "Description: A female makes a speech into a microphone and it is very loud." ,
"role": "assistant", "content": "Answers: (a female makes a speech into a microphone)",
"role": "user", "content": "Description: {new caption}"  ]

messages = [ "role": "user", "content": "'Generate a sentence that contains several different sounds to make a relative whole story, organize them by words indicating time order, don't describe the things unrelevant to sound. First print out the generated sentence and later list the single sound events in the time order they occur. Here are some examples:
Events: honking of a toy trumpet; dog's howl; child's laughter'" ,
"role": "assistant", "content": "Answer: Child's laughter rings out, followed by the obnoxious honking of a toy trumpet. After that, a dog's howl reverberates.
(child's laughter)@(honking of a toy trumpet)@(dog's howl)",
"role": "user", "content": "Events: {new caption} "  ]

