# OpenReview forum: "T2A-Feedback: Improving Basic Capabilities of Text-to-Audio Generation via Fine-grained AI Feedback"
_ICLR.cc/2025/Conference — ICLR 2025 Conference Withdrawn Submission_

### Official Review · Reviewer_rkpB · 2024-10-16

**Soundness:** 2
**Presentation:** 2
**Contribution:** 2
**Rating:** 3
**Confidence:** 5

**Summary:**

This paper introduces an audio preference dataset, named T2A-Feedback, designed to enhance text-to-audio generation systems by improving both prompt adherence and acoustic quality. The dataset leverages three distinct scoring mechanisms: (1) Event Occurrence Score, which assesses the accurate presence of events specified in the prompt; (2) Event Sequence Score, which evaluates the correct sequencing of these events; and (3) Acoustic Harmonic Quality, which measures the overall audio quality based on harmonic consistency. Furthermore, the authors propose a benchmark that evaluates the system's ability to handle long-form captions, including complex multi-event and narrative-driven scenarios.

**Strengths:**

The paper explains all three proposed pipelines in detail and presents corresponding experiments to illustrate the advantage of the proposed scoring metrics.

**Weaknesses:**

1. The concept of using an audio separation model to detect event occurrence is intriguing. However, relying on a CLAP-based separation model to address the limitations of the CLAP model itself seems somewhat unconvincing.
2. The rationale behind determining the event occurrence score by selecting the lowest score needs further clarification.
3. For the event sequence score, identifying the correct sequence based solely on volume levels appears challenging. Additional strategies are warranted, especially for handling events that occur simultaneously.
4. In the event sequence pipeline, the criteria for determining concordant (C) or discordant (D) event pairs are unclear. It’s ambiguous whether only the correct ordering of immediate event pairs is needed or if all events must be in the correct sequence. More explanation and justification for this approach would be beneficial.
5. Only two annotators were involved in scoring the audio sample quality, which raises concerns about the robustness of the experimental results.
6. The paper initially claims that current text-to-audio systems exhibit low performance. However, the authors also used three different text-to-audio systems to generate the proposed T2A-Feedback dataset, which appears somewhat contradictory and requires further justification.
7. The proposed T2A-EpicBench dataset, with an average of 54.8 words per sample, presents a challenge for current text-to-audio generation systems. Given the difficulty in generating long stories within just 10-second audio clips, practical applications are unclear.
8. The experiments for each scoring pipeline, particularly the Acoustic Harmonic Quality (AHQ) evaluation, require further explanation. It is not immediately evident why or how AHQ contributes more effectively.
9. All preference tuning systems discussed in Section 5.2 report higher FAD scores, which warrants additional discussion.
10. Overall, this paper offers an interesting approach to using LLMs for evaluating audio clips and corresponding captions. While the authors attempt to develop a relevant caption dataset and benchmark for their scoring pipelines, the paper lacks sufficient detail and explanation for some experimental aspects. Furthermore, it leans more toward engineering work with limited novelty.

**Questions:**

1. In the event occurrence pipeline, AudioSep is utilized to isolate sub-audio events to facilitate the identification of multiple events through the CLAP model. However, as AudioSep also employs the CLAP model as the text/audio encoder, this approach does not fully circumvent the limitations of CLAP for multi-event scenarios. The authors should further elaborate on the rationale behind this choice and clarify the benefits of combining AudioSep with CLAP in this context.
2. The decision to select the lowest value among all matching scores for event occurrence needs further justification. For example, if there are ten events and only one yields a significantly lower score, this could skew the overall assessment. The authors should validate whether this is an optimal approach for capturing overall performance.
3. Regarding the Event Sequence Score, how does the system handle cases where multiple events occur simultaneously, or when an event spans the entire audio sample? Clarification on these scenarios would strengthen the robustness of the sequence evaluation.
4. In Section 3.3, the authors state that current audio generation models frequently produce low-quality and noisy results (line 247). This claim would benefit from supporting evidence to substantiate it.
5. Further details are needed on the linear predictor mentioned in Section 3.3. Specifically, is this model trained from scratch, or is it fine-tuned from a pre-existing model? A description of its structure would provide better insight.
6. The statement on line 294, “We prompt the LLM with randomly selected events to create varied and natural multi-event audio descriptions,” could use more clarity, especially regarding temporal information. Based on the prompt shown in Appendix C, it appears that the LLM is instructed to assign distinct orders to events. How does the model address cases where audio events actually occur simultaneously?

**Details Of Ethics Concerns:**

No ethic concerns exist.

---

> ### Author Response · Authors · 2024-11-27
> **Response to Reviewer rkpB (1/3)**
>
> ## W1&Q1：Why CLAP-based separation model can address the limitations of CLAP
>
> Firstly, the audio separation model uses CLAP encoder only for encoding conditions (i.e., QueryNet), while the tasks of encoding mixed audio and separating the target audio are handled by the newly learned module (i.e., SeparationNet) of separation model. Therefore, it is difficult to assert that the audio-text alignment in the separation model is inherited from CLAP.
>
> Secondly，CLAP’s limitation in handling multi-event scenarios primarily arises from its inability to effectively comprehend complex, multi-event text descriptions, as shown in the experiments of Sec.5.1. However, when AudioSep is used to isolate sub-audios, **the CLAP encoder only needs to process simple captions containing single events** (segmented by an LLM), which aligns well with CLAP’s strengths. As discussed earlier, **the mixed audio containing multiple events is processed by the SeparationNet within AudioSep, which is specifically trained to encode multi-event audio**. Therefore, **CLAP’s limitation in understanding multi-event audio does not affect the audio separation model’s ability to handle multi-event audio.**
>
> In summary, by leveraging the audio separation model’s capability to process mixed audio and isolate sub-audios based on single-event captions, we can effectively overcome CLAP’s limitations in understanding multi-event audio or text. Furthermore, the superior performance of our pipeline in capturing event occurrences and sequences, compared to CLAP, provides experimental validation of this claim.
>
>
>
> ## W2&Q2：Comparison between selecting lowest or average score for event occurrence score
>
> We tested the effect of selecting the average score and the lowest score among all matching scores for event occurrence judgment as follows:
>
> |               | AudioCaps | Clotho   | MusicCaps |
> | ------------- | --------- | -------- | --------- |
> | EOS (lowest)  | **90.9**  | **90.4** | **99.8**  |
> | EOS (average) | 89.3      | 88.8     | 99.8      |
>
> We find that using the lowest score can better distinguish the caption with extra events for current audio-text datasets. According to the statistical results, we empirically select the lowest score for event occurrence.
>
>
>
> ## W3&Q3: Handling events that occur simultaneously.
>
> By setting volume thresholds on sub-audio, we can identify the occurrence and end times of events. This allows us to use Intersection over Union (IoU) to assess how two events overlap in time, setting a threshold for IoU can filter out the events occur simultaneously, or one event spans the entire audio sample.
>
> The superior F1 score demonstrates the effectiveness of our method in accurately detecting the occurrence and end times of audio events. Furthermore, we collect 800 samples with multiple events occurring simultaneously from the AudioCaps test set. Over these samples, our pipeline achieved an average IoU score of 70.48%, compared to 49.32% from the audio grounding model, underscoring the stronger correlation between the IoU scores detected by our model and the actual instances of simultaneous event occurrences.
>
> ## W4：More justification for concordant or discordant event pairs
>
> When using an LLM to separate different events, we simultaneously instruct it to output the sequence of event occurrences. We then pair these events two by two to form ordered event pairs. For instance, given a sequence derived from the LLM such as E1→E2→E3, the considered ordered event pairs would be （E1→E2），（E2→E3），（E1→E3）. Each event pair is evaluated as either concordant or discordant, and the final Kendall coefficient is calculated accordingly. By fully considering the relationships between all event pairs, our method allows for a more detailed evaluation of event sequences.
>
>
>
> ## W5: More annotators for scoring the audio sample quality
>
> We introduce an additional annotator and analyze the agreements between the learned predictor and the three human annotators (A1, A2, A3) on the acoustic and harmonic test set. The results are summarized below:
>
> |               | A1     | A2     | A3     | Majority |
> | ------------- | ------ | ------ | ------ | -------- |
> | Our predictor | 70.31% | 68.75% | 62.50% | 73.44%   |
> | A1            | -      | 64.06% | 68.75% | 74.33%   |
> | A2            | 64.06% | -      | 65.63% | 71.88%   |
> | A3            | 68.75% | 65.63% | -      | 70.31%   |
>
> Here, A1, A2, and A3 represent the three human annotators. "Majority" indicates the agreement between each annotation (or prediction) and the majority vote of the other three annotators (or predictor).
>
> All the annotators exhibit an agreement rate of over 70% with the majority vote, which demonstrates the reliability of our annotation process.

---

> > ### Author Response · Authors · 2024-11-27
> > **Response to Reviewer rkpB (2/3)**
> >
> > ## W6: Generating preference datasets from weak generative systems
> >
> > The preference data generated by generative systems with relatively low performance and ranked by extra scorers (either human or AI), can further boost the generative model’s performance, which is the basic idea of feedback learning and preference tuning.
> >
> >
> >
> > Due to the inherent randomness in generative systems, the quality of their outputs can vary widely, with some results being surprisingly good and others extremely poor. A high-quality scorer, capable of identifying and ranking both the best and worst results, provides valuable preference signals that guide improvements to the generative model.
> >
> >
> >
> > Similar ideas have been proven effective in LLM, [1] demonstrates a weak language model can be fine-tuned into a stronger one through a self-play approach, where the model generates and annotates preference data only relying on itself. Moreover, our main baseline, Audio-Alpaca, is also collected by ranking the generation results of Tango, and fostering a stronger model, Tango 2.
> >
> > Furthermore, the quantitative results in Tables 4 and 5 of our paper, along with the qualitative improvements showcased on our demo page, further demonstrate the effectiveness of preference tuning.
> >
> >
> >
> > [1] Chen, Zixiang, et al. "Self-Play Fine-Tuning Converts Weak Language Models to Strong Language Models." Forty-first International Conference on Machine Learning.
> >
> >
> >
> > ## W7：10-second audio clips
> >
> > Only generating 10-second audio clips is the limitation of the current audio generation model and seems irrelevant to the focus of our research and can be applied to longer audio. Our scoring pipeline and benchmarks are not constrained to fixed audio lengths.
> >
> >
> >
> >
> >
> > On the other hand, developing generative models that can generate longer harmonic audio is an important research area (although not the focus of our work). For these methods, generating 5 or 6 events harmoniously and accurately in a few dozen seconds or a minute of audio is an important capability with a wide range of application scenarios.
> >
> >
> >
> > ## W8：Further explanation for each scoring pipeline, particularly the Acoustic Harmonic Quality
> >
> > Please refer to the response to Reviewer LA3E’s W1 and W2 for more experiments about event occurrence, event sequence and acoustic harmonic quality evaluation. Refer to the response to Reviewer Reviewer 3Tsj’s W1 for detailed Chi-square tests for our scoring pipelines.
> >
> >
> >
> > ## W9: Negative effect to FAD score
> >
> > Refer to the response to Reviewer Xq56’s Q4.
> >
> >
> >
> > ## W10: Detail and explanation for some experimental aspects
> >
> > We want to clarify that LLM is mainly used to parse text rather than evaluate audio clips and corresponding captions, and the scoring pipeline built based on a series of audio-text basic models such as CLAP, audio separation model, and acoustic harmonic quality predictor is actually used to evaluate the quality of audios and the audio-text alignments.
> >
> > We provide more experiments to exhibit the advantage and effectiveness of each scoring pipeline, as the response to your W8.

---

> > > ### Author Response · Authors · 2024-11-27
> > > **Response to Reviewer rkpB (3/3)**
> > >
> > > ## Q4: Current audio generation models frequently produce low-quality results
> > >
> > > The “before tuning” examples on our demo page illustrate that the existing model may generate audio with poor sound quality. To further support this observation, we provide dozens of examples generated by each model in the updated supplementary materials, which all demonstrate limited sounding quality.
> > >
> > >
> > >
> > > ## Q5: Further details on the linear predictor
> > >
> > > As discussed in Lines 282 and 396–404, the acoustic and harmonic quality predictor is built on the CLAP audio encoder, with an additional simple linear predictor where only the linear predictor is learnable (similar to the aesthetic predictor based on the CLIP image encoder). To provide further insights into training such a subjective predictor, we experimented with different pre-trained audio embeddings, as shown in Table 3 of the paper and summarized below:
> > >
> > > |          | Correlation |
> > > | -------- | ----------- |
> > > | AudioMAE | 0.613       |
> > > | BEAT     | 0.519       |
> > > | CLAP     | **0.786**   |
> > >
> > > Our empirical findings show that CLAP embeddings deliver the best performance.
> > >
> > >
> > >
> > > ## Q6: Events occur simultaneously
> > >
> > > Our work does not specifically investigate the co-occurrence of events in the training data but instead focuses on specifying event orders. However, given the demonstrated strength of our method in determining event co-occurrence (as detailed in our response to W3&Q3), it would be straightforward to incorporate an additional event co-occurrence score, such as the IoU score between events described as occurring simultaneously.
> > >
> > >
> > >
> > > Furthermore, thanks to the improved prompt-following ability enabled by our preference data, we observed enhanced performance in generating simultaneous events, even though our method has not explicitly investigated co-occurrence cases. Experimentally, we selected 800 prompts from the AudioCaps dataset that describe multiple events happening simultaneously (e.g., "Event1 while Event2") and used the IoU score predicted by our pipeline as a metric. A higher score indicates greater alignment between the durations of the two events.
> > >
> > >
> > >
> > > |              |      | IoU        |
> > > | ------------ | ---- | ---------- |
> > > |              | MAA2 | 69.52%     |
> > > | Audio-Alpaca | RAFT | 72.19%     |
> > > |              | DPO  | 67.03%     |
> > > | T2A-Feedback | RAFT | **77.76%** |
> > > |              | DPO  | 74.83%     |
> > >
> > > After preference tuning with T2A-feedback, the audio generation model demonstrates improved capability in generating co-occurring events within the same time period. This unexpected result further underscores the improvement in the model’s basic prompt-following capabilities brought about by our dataset.

---

> > > > ### Comment · Reviewer_rkpB · 2024-11-27
> > > >
> > > > The author did not actually directly answer my questions, for Q5, the most straightforward question is the details of the liner predictor, whether it is trained from scratch or pre-trained. No answer actually appeared in the reply.
> > > > For the reply for Q6, it is more like a suggestion, not the way they try to handle the problem.

---

> > > > > ### Author Response · Authors · 2024-12-02
> > > > > **Further Response to Reviewer rkpB (4/n)**
> > > > >
> > > > > **For Q5**, the linear predictor is initialized randomly.
> > > > >
> > > > >
> > > > >
> > > > > **For Q6**, we provide discussions about the simultaneous occurrence of events, as well as some experiments to prove that our scoring system can evaluate whether events occur simultaneously, and that our current dataset can improve the model's ability to handle simultaneous events.
> > > > >
> > > > >
> > > > >
> > > > > [1] Cui, Ganqu, et al. "ULTRAFEEDBACK: Boosting Language Models with Scaled AI Feedback." Forty-first International Conference on Machine Learning. 2024.
> > > > >
> > > > > [2] Ouyang, Long, et al. "Training language models to follow instructions with human feedback." Advances in neural information processing systems 35 (2022): 27730-27744.
> > > > >
> > > > > [3] Rafailov, Rafael, et al. "Direct preference optimization: Your language model is secretly a reward model." Advances in Neural Information Processing Systems 36 (2024).
> > > > >
> > > > > [4] Burns, Collin, et al. "Weak-to-Strong Generalization: Eliciting Strong Capabilities With Weak Supervision." Forty-first International Conference on Machine Learning.
> > > > >
> > > > > [5] Achiam, Josh, et al. "Gpt-4 technical report." arXiv preprint arXiv:2303.08774 (2023).
> > > > >
> > > > > [6] Kong, Qiuqiang, et al. "Panns: Large-scale pretrained audio neural networks for audio pattern recognition." *IEEE/ACM Transactions on Audio, Speech, and Language Processing* 28 (2020): 2880-2894.
> > > > >
> > > > > [7] https://github.com/LAION-AI/aesthetic-predictor
> > > > >
> > > > >
> > > > >
> > > > > [8] Majumder, Navonil et al. “Tango 2: Aligning Diffusion-based Text-to-Audio Generations through Direct Preference Optimization.” *ArXiv* abs/2404.09956 (2024)
> > > > >
> > > > >
> > > > >
> > > > > [9] Kirstain, Yuval, et al. "Pick-a-pic: An open dataset of user preferences for text-to-image generation." *Advances in Neural Information Processing Systems* 36 (2023): 36652-36663.
> > > > >
> > > > >
> > > > >
> > > > > [10] Zheng, Wendi, et al. "Cogview3: Finer and faster text-to-image generation via relay diffusion." *arXiv preprint arXiv:2403.05121* (2024).

---

> > > ### Comment · Reviewer_rkpB · 2024-11-27
> > >
> > > For W6, the author did not actually solve the concern of using the model that claims to have poor performance to generate audio. If it is the case that the current generation system can generate high-quality models, but just in a random performance, then it is just a simple engineer work proposed in these sections, and the claim that the current model has poor performance is wrong.
> > > For the 10-second concern, I am not asking when only generate a 10-second sample, but concerning just a 10-second audio clip, which cannot actually present a long story that involves more than 50 words. The author did not actually understand and answer my question.
> > > For the rest questions, the author just directly asked me to look into other reviews, which is not a ideal way to answer my concerns.

---

> > > > ### Author Response · Authors · 2024-12-02
> > > > **Further Response to Reviewer rkpB (2/n)**
> > > >
> > > > **For W6,** we first emphasize that ranking the generations of "weak" models as preference data to improve their performance forms the foundation of Reinforcement Learning from Human or AI Feedback (RLHF or RLAIF). This idea has been validated by numerous studies [2][3][4] and systems such as ChatGPT [5]. While the underlying reasons for its effectiveness are beyond the scope of this work, we focus on its downstream application in text-to-audio generation. Our explanation regarding randomly generated better results serves as a simple summary of why feedback learning is effective. A “good” result does not imply perfection in all aspects; often, the model improves in specific dimensions (e.g., better event order but lower audio quality, or high-quality events arranged in suboptimal sequences). Enhancing these foundational dimensions enables the model to produce overall better results.
> > > >
> > > >
> > > >
> > > > We believe that the reviewer's concern lies in the wording and expression. In subsequent versions, we will use more moderate words to illustrate the shortcomings of the current method.
> > > >
> > > >
> > > >
> > > > **Regarding the 10-second concern (W7),** whether a 10-second audio clip can average five events is subjective and beyond the scope of this work. However, we emphasize two key points. First, the limitation of generating only 10-second clips arises from current audio generation models, not our method. Generating longer clips with more events remains a goal for audio generation research, and our approach represents a step toward this long-term objective. Second, the primary limitation of existing methods is not failing to include all events but generating one or two events with very low quality. Our method significantly improves this, which constitutes the primary contribution of our approach.

---

> > > > ### Author Response · Authors · 2024-12-02
> > > > **Further Response to Reviewer rkpB (3/n)**
> > > >
> > > > Regarding the remaining questions, we believe they are similar or even identical to those raised by other reviewers, and our previous responses are trying to address them. We are unsure why our replies to identical concerns (but written in response to others) can not resolve the reviewer’s questions. Nonetheless, as the reviewer insisting, we are happy to reiterate our response here.
> > > >
> > > >
> > > >
> > > > **For Explanation of each scoring pipeline and experiment (W8&W10):**
> > > >
> > > > For the event occurrence (Table 1), we added the state-of-the-art audio tagging model, PANNs [6], as a baseline (matching the top 5 recognized audio categories with open-domain descriptions). We also introduce Clotho (5,225 samples) and MusicCaps (4,434 samples) as two additional benchmarks.
> > > >
> > > >
> > > >
> > > > |      | AudioCaps | Clotho  | MusicCaps |
> > > >
> > > > | ---------- | --------- | -------- | --------- |
> > > >
> > > > | CLAP    | 77.5   | 86.4   | 69.4   |
> > > >
> > > > | PANNs   | 82.0   | 79.9   | 56.1   |
> > > >
> > > > | EOS (ours) | **90.9** | **90.4** | **99.8** |
> > > >
> > > >
> > > >
> > > >
> > > >
> > > > For the event sequence (Table 2), we conducted tests on samples in PicoAudio that contained more than two events (200 samples).
> > > >
> > > >
> > > >
> > > > | method  | Acc   | F1    |
> > > >
> > > > | --------- | -------- | --------- |
> > > >
> > > > | CLAP   | 53.7   | -     |
> > > >
> > > > | PicoAudio | 51.3   | 0.574   |
> > > >
> > > > | ESS_0.1  | 54.2   | **0.606** |
> > > >
> > > > | ESS_0.3  | **57.6** | 0.587   |
> > > >
> > > > | ESS_0.5  | 56.7   | 0.535   |
> > > >
> > > >
> > > >
> > > > The significantly superior performance compared to the two alternative methods highlights our model’s strong understanding of complex event sequences.
> > > >
> > > >
> > > >
> > > > For acoustic and harmonic quality, we scale the audio quality annotation dataset to 2,000 samples and train acoustic and harmonic predictors using more data. The results show that the additional data provides only marginal improvements: a correlation of 0.797 is achieved with 1,600 samples, increasing to 0.803 with 1,800 samples and 0.812 with 2,000 samples. In comparison, the current predictor already achieves a correlation of 0.786.
> > > >
> > > >
> > > >
> > > > We infer that assessing audio quality is a relatively simpler task compared to understanding audio content and high-quality pre-trained audio representations inherently capture substantial information about audio quality.
> > > >
> > > >
> > > >
> > > > A similar phenomenon has been observed in the image domain. For instance, the LAION-Aesthetic-Predictor [7] successfully trained a robust aesthetic scoring model using only 5,000 images, leveraging the strong foundational features provided by CLIP representations.
> > > >
> > > >
> > > >
> > > > Moreover, we conduct chi-square tests on the prediction and human labeling of Event Sequence Score and Acoustic Harmonic Quality. Both tests have 9 degrees of freedom (Event Sequence Score is evenly divided into 4 categories). Based on the data in Tables 2 and 3, the chi-square statistics for event order and audio quality are 88.64 and 133.35, respectively. Using the chi-square distribution table, we find that the p-values for both tests are extremely close to 0. This indicates that, with nearly 100% confidence, the prediction metrics and human labeling metrics are significantly correlated (i.e., reject the null hypothesis).
> > > >
> > > >
> > > >
> > > > **For negative effect to FAD score (W9) :**
> > > >
> > > > FAD and FID estimate the mean and covariance of two sample groups in a high-dimensional feature space and calculate their similarity. A negative correlation between FAD (FID) and subjective metrics is widely observed in the text-to-image and text-to-audio generations. The study Pick-a-Pic [9] for text-to-image feedback learning has discussed this phenomenon, suggesting that it may be correlated to the classifier-free guidance scale mechanism. Larger classifier-free guidance scales tend to produce more vivid samples, which humans generally prefer, but deviate from the distribution of ground truth samples in the test set, resulting in worse (higher) FID (FAD) scores.
> > > >
> > > >
> > > >
> > > > More specifically, this phenomenon is witnessed in Tables 1 and 2 of CogView3[10] (text-to-image method) and Table 3 of Tango2[8] (text-to-audio method), where models achieve higher human preference scores but worse FID (FAD) scores. The negative correlation between FID (FAD) and subjective scores, as consistently shown by previous methods, appears to be an expected outcome when aligning generative models with human preferences.

---

> > ### Comment · Reviewer_rkpB · 2024-11-27
> >
> > Thanks for the reply from the authors.
> > However, the explanation for CLAP model used in AudioSep does not solve my concerns for using a CLAP-based model to handle the limitations of CLAP.
> > Also for the answer for W2/Q2, I am not asking for the performance comparison between using average scoring and lowest scoring, instead, I am asking how to avoid situations where only one yields a significantly lower score. The author did not answer this properly.
> > For W5, there are fewer annotators, and three annotators do not actually solve this concern. In addition, the table is too difficult to read, why are both rows and columns present with A1, A2 and A3?

---

> > > ### Author Response · Authors · 2024-12-02
> > > **Further Response to Reviewer rkpB (1/n)**
> > >
> > > **For W1&Q1,** the audio separation model only using CLAP encoder to encode captions describing single event, and how to handle multi-event and mixed audio is newly learned. The CLAP’s limitation in understanding multi-event captions or audios are naturally not inherited by the audio separation model.
> > >
> > >
> > >
> > > Moreover, beyond theoretical conjectures, we substantiate our claims through comprehensive comparisons with multiple baselines across diverse benchmarks. These experimental results highlight our method's enhanced capability in understanding multi-event audio, which we believe offers a more compelling and practical validation.
> > >
> > >
> > >
> > >
> > >
> > > **For W2 & Q2,** in extreme cases, such as scenarios with ten events where only one event did not occur, our method indeed faces limitations. However, as demonstrated by our experiments, our method statistically achieves superior results on current benchmarks. We argue that addressing the real-world data distribution is more significant than optimizing for extreme edge cases.
> > >
> > >
> > >
> > > **For W5,** we refrain from debating subjective opinions about how many annotations are sufficient. Instead, we demonstrate the reliability of our annotations through a high level of agreement among our annotators. For the readability of our table, we mainly follow the structure of Table 4 in UltraFeedback [1], which indicate ratio of agreements between human annotators and the AI-based scoring pipeline.

---

> ### Comment · Reviewer_rkpB · 2024-11-27
>
> First of all, I would like to thank the authors for their reply. However, I raised several issues in my comments, and although the reply addressed them to some extent, it seems that the authors did not truly resolve my concerns. I'm not sure if this was due to a misunderstanding or some other reason. Moreover, the authors did not appear to respond to my concerns proactively. Many of their answers were either irrelevant to the questions I asked or directed me to look at replies given to others.
>
> Given that there are still many unresolved issues and potential flaws in this paper, I do not intend to raise my score.

---

### Official Review · Reviewer_3Tsj · 2024-10-29

**Soundness:** 3
**Presentation:** 2
**Contribution:** 3
**Rating:** 5
**Confidence:** 4

**Summary:**

This work constitues three metrics and two datasets for text-to-audio (T2A):

* Event occurance score (EOS), Event sequence score (ESS), and Acoustic & harmonic quality (AHQ). EOS and ESS are algorithmic based on off-shelf models, and the final one AHQ is done by training a quality predictor using a manually annotated dataset.

* T2A-Feedback, a 40k captions, 250k generated audio clips with annotated scores.

* T2A-EpicBench, more challenging benchmark captions with longer, story-telling like captions.

T2A-Feedback improves existing T2A models (for example, Make-an-Audio 2) and is claimed to elicit emergent abilities to handle longer text and follow the event order measured by T2A-EpicBench.

**Strengths:**

* A model-based, scalable apparoach to generate large-scale preference dataset is an important direction worth exploring, and T2A-Feedback is one of the early endeavor which warrants credit.

**Weaknesses:**

* For evaluation and dataset papers like this, the authors can consider having more scrutiny in stating significance of the proposed metric's reliability. For example, a chi square test on the confusion matrix and reporting its p-value. Same goes to the benchmarks.

* The scope of verificiation of the proposed metric's robustness is tied to AudioCaps. The readers may question the reliability of the metrics to other audio datasets across different types: Clotho and MusicCaps to name a few.

* The reviewer is not certain about how we should utilize and scale T2A-Feedback because there was no analysis on the impact of the proposed dataset's scale to the model's improved quality. For example, how minimal amount of T2A-Feedback data is needed to elicit such ability? Does the improvement in quality exhibits a ceiling past the scale presented in current experiment? Since T2A-Feedback is one of the first effort in automated preference data generation (which is important direction, granted), I'd like to see deeper insights that can be learned from the proposed method.

**Questions:**

* After the preference tuning (either with Audio-Alpaca or T2A-Feedback), FAD seems to suffer noticeable degradation. As FAD has been considered a go-to metric in T2A community, this may come as a surprise. Can the authors provide explanations and/or analysis on why FAD fails to capture the T2A model's true quality?

---

> ### Author Response · Authors · 2024-11-27
> **Response to Reviewer 3Tsj**
>
> ## W1：Chi-square tests
>
> We conduct chi-square tests on the prediction and human labeling of Event Sequence Score and Acoustic Harmonic Quality.
>
> Both tests have 9 degrees of freedom (Event Sequence Score is evenly divided into 4 categories). Based on the data in Tables 2 and 3, the chi-square statistics for event order and audio quality are 88.64 and 133.35, respectively. Using the chi-square distribution table, we find that the p-values for both tests are extremely close to 0. This indicates that, with nearly 100% confidence, the prediction metrics and human labeling metrics are significantly correlated (i.e., reject the null hypothesis).
>
>
>
> ## W2：Experiments on more datasets
>
> We compare our methods with more baselines on more benchmarks, refer to the response to Reviewer LA3E’s W1.
>
> ## W3：Impact of dataset's scale to the model's improved quality
>
> We tried to use different proportions of training data for preference tuning as follows:
>
> | Data proportion | FAD(↓)   | KL (↓)   | IS(↑)     | CLAP(↑)   | EOS. (↑)  | ESS. (↑) | AQ. (↑)  |
> | --------------- | -------- | -------- | --------- | --------- | --------- | -------- | -------- |
> | 0%              | **1.82** | 1.44     | 10.03     | 69.97     | 42.05     | 0.53     | 2.33     |
> | 20%             | 3.44     | 1.53     | 10.18     | 69.81     | 43.78     | 0.51     | 2.27     |
> | 40%             | 2.63     | 1.40     | 10.99     | 72.24     | 46.31     | 0.52     | 2.44     |
> | 60%             | 2.39     | 1.34     | 11.20     | 73.59     | 47.72     | 0.57     | 2.54     |
> | 80%             | 2.63     | 1.38     | 11.18     | 73.00     | 49.00     | 0.55     | 2.50     |
> | 100%            | 2.64     | **1.31** | **11.35** | **74.00** | **49.58** | **0.57** | **2.57** |
>
> The results indicate that utilizing the entire training dataset yields the best outcomes, underscoring the effectiveness of our preference data annotation process and its scalability potential.
>
>
>
> ## Q1：Negative effect to FAD score
>
> Please refer to the response to Reviewer Xq56’s Q4.

---

> ### Author Response · Authors · 2024-12-02
> **Looking forward to your feedback!**
>
> Dear Reviewer 3Tsj,
>
> Thank you for your valuable comments and suggestions, which have been very helpful to us. We have carefully addressed the concerns raised in your reviews and included additional experimental results.
>
> As the discussion deadline is approaching, we would greatly appreciate it if you could take some time to provide further feedback on whether our responses adequately address your concerns.
>
> Best regards,
>
> The Authors

---

### Official Review · Reviewer_Xq56 · 2024-11-01

**Soundness:** 3
**Presentation:** 3
**Contribution:** 2
**Rating:** 6
**Confidence:** 5

**Summary:**

This paper addresses the challenge of aligning Text-to-Audio generation models with human preferences, especially for complex audio prompts involving multiple events. The authors first introduce three scoring pipelines, Event Occurrence Score, Event Sequence Score, and Acoustic Harmonic Quality, that measure the basic capabilities of TTA generation. Then construct a T2A-Feedback dataset. Also, a benchmark T2A-EpicBench is derived to evaluate the advanced capabilities of T2A models. In the experimental section, the paper first verifies the effectiveness of the proposed evaluation measures and then uses the constructed dataset for preference tuning, thereby enhancing the performance of state-of-the-art models.

**Strengths:**

This paper is well-formulated and clear in structure, starting from the current deficiencies in TTA models, such as event occurrence, sequence prompt-following, and quality issues. It then progresses to the construction of the scoring pipelines, the introduction of the dataset and benchmark, and finally, demonstrates how preference-tuning with the dataset improves model performance. Each AI feedback scoring metric is described in detail, making it easy to replicate. The clear presentation of quantitative and qualitative analyses helps intuitively showcase the effectiveness of these innovations in enhancing the T2A model.

**Weaknesses:**

1.	The paper does not mention the impact of the validation dataset on other models, such as AudioLDM 2 or Tango 2, to ensure the dataset’s generalizability. Additionally, the benchmark has not been tested on other models, making it difficult to determine the benchmark’s discriminative power and effectiveness.
2.	Lines 513 to 515 lack further analysis on why the model performs well in T2A-EpicBench’s long-text scenarios, despite T2A-Feedback focusing more on short-text and single-event descriptions.

**Questions:**

1.	In lines 223-225, could you explain how, after determining the threshold value, the separated event’s occurrence time in the original audio is identified?
	2.	In line 372, the paper is positioned to address the sequence prompt-following issue for complex audio prompts involving multiple events. Were experiments conducted on samples with more than two events?
	3.	In lines 428-429, why were DPO and RAFT chosen for preference tuning instead of other preference tuning methods (such as PPO)?
	4.	Could you analyze whether the increase in FAD after fine-tuning in Table 4 contradicts the improvement in the acoustic quality metric?

---

> ### Author Response · Authors · 2024-11-27
> **Response to Reviewer Xq56**
>
> ## W1：The impact of our benchmark on other models
>
> The performance of AudioLDM 2 and Tango 2 on EpicBench is as follows:
>
> |                                    | EOS       | ESS      | AHQ      |
> | ---------------------------------- | --------- | -------- | -------- |
> | Make-an-audio 2                    | 14.21     | 0.03     | 1.96     |
> | AudioLDM2                          | 16.35     | 0.04     | 1.98     |
> | Tango2                             | 19.42     | 0.07     | **2.11** |
> | Make an audio 2+T2A-FeedBack (DPO) | **19.96** | **0.13** | 2.10     |
>
> The improvements observed across Make-an-audio 2, AudioLDM2, and Tango2 on EpicBench align with their inherent capabilities, with newer and more advanced models achieving better results. This indirectly validates the robustness and effectiveness of our benchmark and AI-based scoring pipeline.
>
> Moreover, we observed that although the Make-an-audio 2 model does not perform well on EpicBench initially, it achieves the best performance after feedback alignment with T2A-Feedback. This highlights the practicality and significance of our dataset.
>
>
>
> ## W2：Why T2A-Feedback focused on short caption can improve the performance on T2A-EpicBench
>
> Although our dataset does not contain any long-caption data, it contains preference data across three dimensions: event occurrence, event sequence, and acoustic quality. By improving the model’s foundational capabilities in these areas, it achieves emergent improvements in long, narrative, and complex scenarios. This exciting phenomenon further highlights the value of preference tuning for audio generation.
>
>
>
> ## Q1：Identification of separated event’s occurrence time
>
> First, we normalize the audio volume to a range of 0–1. The event occurrence time is defined as the first moment the normalized volume exceeds the threshold (0.3 in our default setting), and the event end time corresponds to the last timestamp that the normalized volume exceeds the threshold.
>
>
> ## Q2：Event sequence experiments on samples with more than two events
>
> We conducted tests on samples in PicoAudio that contain more than two events (200 in total) .
>
> | method    | Acc      | F1        |
> | --------- | -------- | --------- |
> | CLAP   | 53.7     | -         |
> | PicoAudio | 51.3     | 0.574     |
> | ESS_0.1   | 54.2     | **0.606**     |
> | ESS_0.3   | **57.6** | 0.587 |
> | ESS_0.5   | 56.7    | 0.535    |
>
> The significantly superior performance compared to the two alternative methods highlights our AI-based scoring pipeline’s strong understanding of complex event sequences.
>
>
>
> ## Q3: Why choose DPO and RAFT for preference tuning, and results with PPO
>
> The primary reason we selected DPO and RAFT for preference tuning is that these methods have been extensively applied to diffusion models and come with stable, readily available open-source implementations. Furthermore, prior work on audio and image preference datasets has utilized only one of these approaches—Tango 2 [1] adopts DPO, while RichHF-18K [2] relies on RAFT. We are currently experimenting with PPO strategies for feedback learning, and the results will be updated soon.
>
> ## Q4: Negative effect to FAD score
>
> FAD and FID estimate the mean and covariance of two sample groups in a high-dimensional feature space and calculate their similarity. A negative correlation between FAD (FID) and subjective metrics is widely observed in the text-to-image and text-to-audio generations. The study Pick-a-Pic [3] for text-to-image feedback learning has discussed this phenomenon, suggesting that it may be correlated to the classifier-free guidance scale mechanism. Larger classifier-free guidance scales tend to produce more vivid samples, which humans generally prefer, but deviate from the distribution of ground truth samples in the test set, resulting in worse (higher) FID (FAD) scores.
>
> More specifically, this phenomenon is witnessed in Tables 1 and 2 of CogView3[4] (text-to-image method) and Table 3 of Tango2[1] (text-to-audio method), where models achieve higher human preference scores but worse FID (FAD) scores. The negative correlation between FID (FAD) and subjective scores, as consistently shown by previous methods, appears to be an expected outcome when aligning generative models with human preferences.
>
> [1] Majumder, Navonil et al. “Tango 2: Aligning Diffusion-based Text-to-Audio Generations through Direct Preference Optimization.” *ArXiv* abs/2404.09956 (2024)
>
> [2] Liang, Youwei et al. “Rich Human Feedback for Text-to-Image Generation.” *2024 IEEE/CVF Conference on Computer Vision and Pattern Recognition (CVPR)* (2023): 19401-19411.
>
> [3] Kirstain, Yuval, et al. "Pick-a-pic: An open dataset of user preferences for text-to-image generation." *Advances in Neural Information Processing Systems* 36 (2023): 36652-36663.
>
> [4] Zheng, Wendi, et al. "Cogview3: Finer and faster text-to-image generation via relay diffusion." *arXiv preprint arXiv:2403.05121* (2024).

---

> > ### Author Response · Authors · 2024-12-02
> > **Looking forward to your feedback!**
> >
> > Dear Reviewer Xq56,
> >
> > Thank you for your valuable comments and suggestions, which have been very helpful to us. We have carefully addressed the concerns raised in your reviews and included additional experimental results. Unfortunately, we have not yet been able to port PPO to our codebase these days, but we are actively working on this and will provide updated results in a future version.
> >
> > As the discussion deadline approaches, we kindly ask if you review our responses and let us know whether they sufficiently address your concerns.
> >
> > Best regards,
> >
> > The Authors

---

### Official Review · Reviewer_LA3E · 2024-11-03

**Soundness:** 3
**Presentation:** 3
**Contribution:** 3
**Rating:** 6
**Confidence:** 5

**Summary:**

This paper focuses on constructing a preference feedback dataset for Text to Audio (T2A). It proposes a fine-grained scoring pipeline that relies on both AI models and human annotations to collaboratively establish an evaluation metric. This metric assesses both prompt-following (including event occurrence and sequence) as well as acoustic and harmonic quality. Based on this framework, the authors have developed the T2A-Feedback dataset. Additionally, they introduce T2A-EpicBench, a challenging set of evaluation samples.

**Strengths:**

1. The motivation presented in the paper is very clear. Effectively evaluating the generation quality of TTA, aligning generated audio with human perceptual systems, and designing reliable evaluation sets are all crucial issues in the TTA and audio generation fields.

2. The methodology is reasonable. Event occurrence and sequence, as well as acoustics and harmonic quality, are indeed three important dimensions in the TTA problem. Utilizing AI models to assess event occurrence and sequence, while employing human scoring for acoustic and harmonic quality evaluation, creates a hybrid assessment system that is robust and lends itself to automated scaling.

3. The experimental results are good. The results of preference alignment and the samples on the demo page demonstrate the effectiveness of the T2A-Feedback dataset.

**Weaknesses:**

1. My primary concern pertains to the scoring pipeline for event occurrence and sequence. In the current design, audio source separation is a critical component. From my experience, audio events in TTA datasets are often quite mixed, with multiple events potentially occurring simultaneously. The existing source separation models seem to struggle with effectively isolating various events. Furthermore, these separated results need to be accurately matched with the multiple event descriptions generated by the large language model (LLM). The authors do not appear to showcase any examples of audio separation, nor do they demonstrate how these separations match with the multiple captions generated by the large language model (LLM) on the demo website or in the supplementary materials. Additionally, the experimental setups presented in Tables 1 and 2 are overly simplistic; for instance, Table 1 only compares CLAP, and Table 2 utilizes samples with only two events. This lack of complexity seems insufficient to substantiate the reliability of the AI models currently in use.

2. Regarding the assessment of acoustic and harmonic quality, the authors suggest that only 1,000 labeled samples are necessary to train a satisfactory subjective quality predictor. Intuitively, this number seems too small. In such a limited training/testing experiment, even if the predictor performs well, it may simply be due to the overly simplistic distribution of the existing TTA dataset.

3. Concerning practical applicability in real-world scenarios, I acknowledge that the alignment learning via T2A-Feedback has led to some improvements in model performance as observed on the demo website. However, the overall quality of the models remains inadequate and far from sufficient to support applications in real-world settings.

**Questions:**

I hope to see a response during the rebuttal phase addressing my concerns regarding the reliability of the event separation and event sequence produced by the AI model. Specifically, I would like to see more concrete qualitative or quantitative results. If the authors can adequately address this issue, I would be willing to reconsider my rating to accept.

---

> ### Author Response · Authors · 2024-11-27
> **Response to Reviewer LA3E**
>
> ## W1: More results on event occurrence and sequence
>
> In the updated supplementary materials, we provide some examples of the audio separation model on mixed audios. We believe that the quality of a model is relative. While existing audio separation models may perform poorly in certain cases, our empirical observations suggest that the pipeline built upon audio separation model still outperform alternative approaches in tasks such as identifying the occurrence of events and their sequences.
>
>
> For the event occurrence experiments (Table 1), we added the state-of-the-art audio tagging model, PANNs [1], as a baseline (matching the top 5 recognized audio categories with open-domain descriptions). We also introduce Clotho (5,225 samples) and MusicCaps (4,434 samples) as two additional benchmarks.
>
> |            | AudioCaps | Clotho   | MusicCaps |
> | ---------- | --------- | -------- | --------- |
> | CLAP       | 77.5      | 86.4     | 69.4      |
> | PANNs      | 82.0      | 79.9     | 56.1      |
> | EOS (ours) | **90.9**  | **90.4** | **99.8**  |
>
>
> For the event sequence experiments (Table 2), we conducted tests on samples in PicoAudio that contained more than two events (200 samples).
>
> | method    | Acc      | F1        |
> | --------- | -------- | --------- |
> | CLAP   | 53.7     | -         |
> | PicoAudio | 51.3     | 0.574     |
> | ESS_0.1   | 54.2     | **0.606**     |
> | ESS_0.3   | **57.6** | 0.587 |
> | ESS_0.5   | 56.7    | 0.535    |
>
> The significantly superior performance compared to the two alternative methods highlights our model’s strong understanding of complex event sequences.
>
> [1] Kong, Qiuqiang, et al. "Panns: Large-scale pretrained audio neural networks for audio pattern recognition." *IEEE/ACM Transactions on Audio, Speech, and Language Processing* 28 (2020): 2880-2894.
>
>
>
> ## W2: More results on acoustic and harmonic quality
>
> We scale the audio quality annotation dataset to 2,000 samples and train acoustic and harmonic predictors using more data. The results show that the additional data provides only marginal improvements: a correlation of 0.797 is achieved with 1,600 samples, increasing to 0.803 with 1,800 samples and 0.812 with 2,000 samples. In comparison, the current predictor already achieves a correlation of 0.786.
>
> We infer that assessing audio quality is a relatively simpler task compared to understanding audio content and high-quality pre-trained audio representations inherently capture substantial information about audio quality.
>
> A similar phenomenon has been observed in the image domain. For instance, the LAION-Aesthetic-Predictor [2] successfully trained a robust aesthetic scoring model using only 5,000 images, leveraging the strong foundational features provided by CLIP representations.
>
> [2] https://github.com/LAION-AI/aesthetic-predictor
>
> ## W3: Overall quality of the models remains inadequate
>
> Automatically generating high-quality and harmonious audio from detailed, narrative, and multi-event scenarios remains a long-term goal. The performance of the audio generation model depends on both the pre-training phase and the post-training phase (fine-tuning and feedback learning). To fully address the challenge of generating coherent audio for long narrative prompts, improvements are needed across the entire process.

---

> ### Comment · Reviewer_LA3E · 2024-12-01
> **Increase rating from 5 to 6**
>
> Thanks for your additional supplementary materials and experimental results, which addressed most of my concerns. I have increased the rating from 5 to 6. I hope you can open-source everything if the paper is accepted.

---

> ### Author Response · Authors · 2024-12-02
> **Thank you for your support!**
>
> Dear Reviewer LA3E,
>
> We appreciate your kind support! In our final revision, we will further improve the paper by incorporating the valuable insights gained from the rebuttal discussions. Thank you again!
>
> Best regards,
>
> The Authors

---

### Note · Authors · 2024-12-13

I have read and agree with the venue's withdrawal policy on behalf of myself and my co-authors.